# Diclofenac Potentiates Sorafenib-Based Treatments of Hepatocellular Carcinoma by Enhancing Oxidative Stress

**DOI:** 10.3390/cancers11101453

**Published:** 2019-09-27

**Authors:** Adrian Paul Duval, Laetitia Troquier, Olga de Souza Silva, Nicolas Demartines, Olivier Dormond

**Affiliations:** Department of Visceral Surgery, Lausanne University Hospital and University of Lausanne, Av de Beaumont, 1011 Lausanne, Switzerland; adrian.duval@chuv.ch (A.P.D.); laetitia.troquier@chuv.ch (L.T.); olga.de-souza-silva@chuv.ch (O.d.S.S.); demartines@chuv.ch (N.D.)

**Keywords:** sorafenib, hepatocellular carcinoma, diclofenac, oxidative stress, cell death, co-therapy

## Abstract

Sorafenib is the first developed systemic treatment for advanced forms of hepatocellular carcinoma, which constitutes the most frequent form of primary liver cancers and is a major global health burden. Although statistically significant, the positive effect of sorafenib on median survival remains modest, highlighting the need to develop novel therapeutic approaches. In this report, we introduce diclofenac, a nonsteroidal anti-inflammatory drug, as a potent catalyzer of sorafenib anticancer efficacy. Treatment of three different hepatocellular cancer cells (Huh-7, HepG2, and PLC-PRF-5) with sorafenib (5 µM, 24 h) and diclofenac (100 µM, 24 h) significantly increased cancer cell death compared to sorafenib or diclofenac alone. Anti-oxidant compounds, including N-acetyl-cysteine and ascorbic acid, reversed the deleterious effects of diclofenac/sorafenib co-therapy, suggesting that the generation of toxic levels of oxidative stress was responsible for cell death. Accordingly, whereas diclofenac increased production of mitochondrial oxygen reactive species, sorafenib decreased concentrations of glutathione. We further show that tumor burden was significantly diminished in mice bearing tumor xenografts following sorafenib/diclofenac co-therapy when compared to sorafenib or diclofenac alone. Taken together, these results highlight the anticancer benefits of sorafenib/diclofenac co-therapy in hepatocellular carcinoma. They further indicate that combining sorafenib with compounds that increase oxidative stress represents a valuable treatment strategy in hepatocellular carcinoma.

## 1. Introduction

Hepatocellular carcinoma (HCC) constitutes 75–95% of primary liver cancers, and was the sixth most commonly diagnosed and fourth most deadly cancer in 2018 [1]. Due to an increasing incidence, exceedingly aggressive nature, late-stage symptomatic manifestation, and poor survival rate, HCC represents an important global health burden. Limited therapeutic options exist for advanced forms of HCC, highlighting a need to develop new therapeutic approaches [2].

Sorafenib is a multi-tyrosine-kinase inhibitor whose targets include platelet-derived growth factor receptor β, vascular endothelial growth factor receptor (VEGFR)-1, VEGFR2, VEGFR3, Fms-like tyrosine kinase 3 (flt-3), c-KIT, and rearranged during transfection kinase (RET). In addition, it inhibits the serine/threonine kinase Raf-1 [3]. Sorafenib was the first approved systemic treatment for advanced HCC, augmenting median survival by three months [4]. It is thus critical to find novel and quickly available methods to improve its efficiency. Recent research on sorafenib showed that it affects oxidative homeostasis of cancer cells by increasing reactive oxygen species (ROS) production [5]. In addition, sorafenib reduces cellular antioxidant defenses by inhibiting system x_c_^-^, a cystine/glutamate antiporter that provides the cysteine necessary for the synthesis of glutathione (GSH) [6].

ROS are chemically active molecules that perform crucial functions in living organisms [7]. Indeed, a moderate accumulation of ROS promotes differentiation and proliferation, whereas inordinate quantities of ROS cause oxidative damage to DNA, lipids, and proteins that trigger cell death [8]. Accordingly, increasing ROS production to toxic levels represents a treatment strategy in cancer [9]. Several chemotherapeutic drugs and radiotherapy do in fact exercise their cytotoxic effects through the generation of ROS [10].

In this study, we hypothesized that since sorafenib increases oxidative stress, combining it with agents that increase intracellular ROS would result in enhanced cancer cell death. To probe this, we used diclofenac, a commonly prescribed nonsteroidal anti-inflammatory drug (NSAID), known to generate ROS by altering mitochondrial function [11], in combination with sorafenib to test their anticancer combination potential in HCC cell lines in vitro and in vivo. The anticancer potential of the combined diclofenac/sorafenib treatment is further supported by the synergistic cytotoxicity of both drugs observed in melanoma cells, which therefore needs to be investigated in the context of HCC [12].

## 2. Results

### 2.1. Sorafenib and Diclofenac Co-Therapy Increases HCC Cell Death

Three HCC cell lines of human origin (Huh-7, HepG2, and PLC-PRF-5) were submitted to 24 or 48 h of treatment with sorafenib (5 μM), diclofenac (100 μM), or co-therapy. Concentrations of 5 μM sorafenib and 100 μM diclofenac were chosen, as previous studies showed their anticancer efficacy [5,13]. After 24 or 48 h of treatment, MTS proliferation assay results showed a significant decrease in cell proliferation in Huh-7, HepG2, and PLC-PRF-5 cells treated with sorafenib combined with diclofenac compared to either drug applied alone (Figure 1A). We further tested different concentrations of both sorafenib and diclofenac, and found a direct dose- and time-dependent negative effect on cellular proliferation (Appendix A). These results were confirmed by microscopic observation of crystal violet-stained cells, which showed a decrease in cell density when the co-therapy was applied (Figure 1B). Flow cytometry cell cycle analysis with propidium iodide staining showed a significant increase in the hypodiploid fraction, supporting the hypothesis that sorafenib and diclofenac co-therapy induced cell death (Figure 1C). Taken together, these results show that sorafenib, combined with diclofenac, induces HCC cell death more efficiently.

### 2.2. Sorafenib and Diclofenac Increase Oxidative Stress in HCC Cells

Previous studies showed that both sorafenib and diclofenac induce oxidative stress [11]. To test oxidative stress levels in HCC cells exposed to sorafenib and diclofenac, we determined intracellular ROS levels. Diclofenac significantly increased ROS levels after 5 h of treatment in all three HCC cell lines tested (Figure 2a). In contrast, after 5 h, sorafenib had no significant effect on ROS levels, and combining sorafenib with diclofenac did not increase ROS levels compared to diclofenac alone. Decreasing anti-oxidant defenses also contributes to oxidative stress generation. In this context, we determined total glutathione levels, the most abundant antioxidant in cells, in HCC cell lines after treatment with diclofenac and sorafenib. We found that only sorafenib significantly reduced GSH quantities, and not diclofenac (Figure 2b). Together with total GSH quantity, the ratio of reduced GSH to oxidized GSH (GSSG) reflects the oxidative stress. We observed that sorafenib, in combination with diclofenac, significantly decreased the GSH/GSSG ratio compared to either treatment alone or to the control (Figure 2b). Taken together, these experiments show that sorafenib/diclofenac co-therapy increases oxidative stress in HCC.

### 2.3. Blocking Oxidative Stress Prevents Sorafenib/Diclofenac-Mediated HCC Cell Death

We investigated the role of oxidative stress in sorafenib/diclofenac-induced HCC cell death by treating HCC cells with the anti-oxidant N-acetyl-cysteine (NAC) concomitantly with sorafenib and diclofenac [14]. N-acetyl-alanine (NAA) was used as a control. We found that NAC significantly reduced ROS levels generated by diclofenac or diclofenac/sorafenib co-therapy, whereas NAA had no effect (Figure 3a). Furthermore, NAC significantly increased HCC cell growth in the sorafenib/diclofenac treatment condition (Figure 3b). Cell cycle analysis revealed that NAC protected HCC cells from sorafenib/diclofenac-induced cell death (Figure 3c). Conversely, NAA had no effect. Together with NAC, we also tested the effect of the anti-oxidant ascorbic acid (AA) in protecting cells from sorafenib/diclofenac-induced HCC cell death [15]. As for NAC, AA significantly increased HCC cell growth when treated with sorafenib/diclofenac (Figure 3d). High levels of ROS are known triggers of several death processes including apoptosis, autophagy-mediated cell death, and/or necroptosis [16]. We used inhibitors of these pathways to test their involvement in sorafenib/diclofenac-induced HCC cell death. However, neither Z-VAD-FMK, chloroquine, nor necrostatin-1, inhibitors of apoptosis, autophagy and necroptosis respectively, protected HCC cells from sorafenib/diclofenac-induced cell death (Supplemental Appendix A).

### 2.4. Increased Mitochondrial ROS in HCC by Sorafenib/Diclofenac

Mitochondrial respiration represents a major source of intracellular ROS [17]. We therefore probed the role of mitochondria as a source of ROS in HCC cells exposed to sorafenib/diclofenac. Mitochondrial ROS was significantly increased by sorafenib/diclofenac co-therapy in HCC cells (Figure 4a). Diclofenac or sorafenib already significantly augmented mitochondrial ROS levels in HepG2 and PLC-PRF-5 cells, but not in Huh-7 cells (Figure 4a). To characterize the role of mitochondrial ROS in sorafenib/diclofenac-mediated HCC cell death, we generated mitochondrial DNA-deficient cancer cells, as mitochondrial ROS production by such cells is significantly reduced [18]. Huh-7 mitochondrial DNA-deficient cells (Huh-7 MD) were produced by exposing Huh-7 cells to ethidium bromide for 12 weeks [18]. Mitochondrial ROS levels were significantly decreased in Huh-7 MD cells compared to Huh-7 cells. In addition, mitochondrial ROS levels were markedly reduced following co-therapy with sorafenib/diclofenac in Huh-7 MD cells compared to Huh-7 cells (Figure 4b). We further observed, by MTS proliferation assay, that the inhibition of HCC cell growth, mediated by sorafenib/diclofenac, was significantly decreased in Huh-7 MD compared with Huh-7 cells (Figure 4c). Taken together, these results indicate that increased mitochondrial ROS levels induced by sorafenib/diclofenac cotreatment contribute to sorafenib/diclofenac-mediated HCC cell death.

### 2.5. Diclofenac Potentiates the Anticancer Efficacy of Sorafenib In Vivo

We tested the anticancer efficacy of diclofenac/sorafenib co-therapy in nude mice bearing Huh-7 tumor xenografts. We found that diclofenac, in combination with sorafenib, significantly reduced tumor growth compared to untreated mice or mice treated with either diclofenac or sorafenib (Figure 5a). At sacrifice, mean tumor volumes were significantly reduced by diclofenac/sorafenib co-therapy compared to diclofenac or sorafenib alone or untreated mice (Figure 5b). Similar findings were found for mean tumor weights except for diclofenac, where mean tumor weight was not statistically significantly reduced compared to untreated control cases (Figure 5c). No significant difference in mice weights at sacrifice were monitored between the different treatment groups, indicating that treatments were well tolerated by mice (Figure 5d).

## 3. Discussion

Development of therapeutic strategies to improve the outcome of advanced HCC is necessary as approved therapies for this disease induce only a modest improvement in the median overall survival rate [19]. In this study, we report that combining sorafenib, the first approved drug for advanced HCC, with diclofenac provides stronger anticancer efficacy than either drug alone. This effect results from the augmented cell death induced by increased oxidative stress in vitro.

Several studies have linked sorafenib to oxidative stress. In hepatic cell lines, sorafenib increased the production of ROS via stimulation of nicotinamide adenine dinucleotide phosphate (NADPH) oxidases [5]. The superoxide dismutase mimic MnTBAP partially reversed the cytotoxic effects of sorafenib, highlighting the role of the superoxide anion in this process. More importantly, the effect of ROS on protein oxidation was detected in HCC patients treated with sorafenib and predicted drug effectiveness, suggesting that oxidative stress plays an important role in the anticancer effect of sorafenib [5]. Besides increasing production of ROS, sorafenib further reduces antioxidant defense mechanisms. In particular, inhibition of system x_c_^-^ by sorafenib (2.5–10 µM) resulted in decreased importation of cystine and, consequently, GSH synthesis [6]. Consistent with these findings, we observed that the treatment of HCC cell lines with sorafenib reduced GSH levels (Figure 2b). The importance of oxidative stress in mediating the anticancer effect of sorafenib is further supported by the observation that upregulation of the expression of the nuclear factor erythroid 2-related factor 2, a key regulator of the antioxidant response, confers resistance to sorafenib [20].

Our work emphasizes the effects of increasing oxidative stress to improve the anticancer efficacy of sorafenib. Indeed, we observed that diclofenac/sorafenib-induced cell death was prevented by anti-oxidant agents (Figure 3). Previous studies, using a similar concentration of sorafenib as in our study, support this hypothesis. For instance, the combination of sorafenib with tetrandrine exhibited synergistic anti-tumor activity that relied on the generation of toxic levels of intracellular ROS [21]. Similarly, sorafenib combined with oleanolic acid displayed increased anti-tumor effects compared to either drug alone due to ROS-mediated cell death [22]. Furthermore, aspirin and sorafenib together were seen to potentiate the cytotoxicity of cisplatin in head and neck cancer cells via inhibition of system x_c_^-^ [23].

We chose diclofenac as previous reports indicated its ability to increase mitochondrial ROS by alteration of mitochondrial function [24]. Similarly, we found increased mitochondrial ROS formation following the treatment of HCC cancer cells with diclofenac (Figure 2a). Increased ROS production is not limited to diclofenac among NSAIDs. For instance, indomethacin enhances mitochondrial ROS production in gastric cancer cells [25]. Sulindac and its metabolites sulindac sulfide and sulindac sulfone can stimulate ROS generation resulting in oxidative DNA damage in several cancer cell lines [26,27]. Notably, conflicting reports exist regarding NSAID-mediated oxidative stress. In vitro, scavenging activity for hydrogen peroxide by NSAIDs was demonstrated [28]. Similarly, sulindac and its metabolites exhibit scavenging activity against ROS [29]. In addition, increased activity of antioxidant systems, such as superoxide dismutase or glutathione peroxidase by NSAID, was reported as part of a biological response to the initial induction of ROS by NSAID [30,31]. This suggests that the ability of NSAIDs to increase oxidative stress should be carefully tested before using them as pro-oxidant agents. Several studies have demonstrated the anticancer activity of NSAIDs. In particular, epidemiologic evidence shows that long-term use of NSAIDs is associated with a reduced incidence of cancer [32]. The molecular mechanisms underlying the anti-neoplastic effects of NSAIDs have been partially identified. In particular, mechanisms independent of cyclooxygenase (COX) inhibition are also involved as NSAID metabolites that do not inhibit COX retain their anticancer effect [33]. Similarly, it was suggested that NSAID-mediated ROS production might be COX-independent [31], as sulindac sulfone, a metabolite of sulindac, has no COX-inhibitory activity but is a strong inducer of ROS. In contrast, NS-398, a selective COX-2 inhibitor, only weakly induces ROS production [31].

Our results highlight the potential to use diclofenac in combination with sorafenib in HCC. Consistent with our findings, the additive effect of various NSAIDs in combination with sorafenib was demonstrated in vitro in the HCC cell line HepG2 [34]. Of note, the efficacy of sorafenib/diclofenac combination treatment is not limited to HCC, as a synthetic lethal screening with small molecule inhibitors demonstrated a synergistic cytotoxicity between sorafenib and diclofenac in melanoma cell lines [12]. Downregulation of survival-related genes was induced by the treatment combination. Interestingly, among three NSAIDs tested in combination with sorafenib, diclofenac displayed the greatest anticancer efficacy.

Translating our treatment regimen to patients will require careful assessments of drug toxicities. Indeed hepatotoxicity due to NSAID-induced hepatocellular injury has been well documented [35]. Although the incidence of liver injury induced by NSAIDs remains low, NSAIDs could be particularly detrimental in patients with HCC as they frequently present liver cirrhosis. In addition, NSAIDs could precipitate renal failure and increase risk of bleeding in these patients. Besides its own liver toxicity, diclofenac interaction with sorafenib might induce further side effects. In fact, the metabolism of diclofenac might be slowed down via sorafenib-mediated inhibition of the cytochrome P450 2C9 metabolic pathway of diclofenac [36,37]. Consistent with this, acute liver failure was reported in a patient suffering from kidney cancer who was treated with sorafenib and diclofenac simultaneously [38]. Hence, caution has to be taken before translating such a treatment strategy in the clinic, and treatment should be initially tested in HCC patients without cirrhosis to minimize the risk of developing toxic side effects.

As for other targeted therapies, development of resistance is a major obstacle to the anticancer benefit of sorafenib [39]. In this context, overexpression and activation of the transcription factor c-jun plays an important role in contributing to sorafenib resistance [40,41,42]. Indeed, activation of c-jun is enhanced by sorafenib in HCC cell lines, and inhibition of c-jun increases sorafenib-mediated apoptosis [40]. Importantly, in HCC patients, tumors with high levels of c-jun phosphorylation are less sensitive to sorafenib compared to tumors with low levels [41]. Therefore, it will be important to test in future experiments our treatment strategy in the context of sorafenib resistance, and particularly address the effects of sorafenib/diclofenac on c-jun activity. Of note, it was reported that diclofenac increases c-jun expression in acute myeloid leukemia cell lines [43].

Resistance to sorafenib is also exerted by the expression of transporters able to export sorafenib, thus reducing intracellular concentration of sorafenib [44]. Interestingly, β-caryophyllene oxide, a natural component of many essential oils, is able to reduce sorafenib efflux from tumor cells [45]. Consequently, it increases the antiproliferative efficacy of sorafenib in vitro and in vivo. It is therefore worth investigating whether β-caryophyllene oxide could also potentiate the efficacy of sorafenib/diclofenac co-therapy.

Despite the use of inhibitors of cell death pathways, we were not able to identify the precise mechanism responsible for sorafenib/diclofenac-mediated cell death (Appendix A). Similarly, histological analysis of tumor xenografts did not demonstrate reduced cell proliferation or increased apoptosis (data not shown). Hence, additional experiments are needed to dissect the mechanisms that lead to cell death following increased oxidative stress. Furthermore, in tumor xenografts, the effect of sorafenib/diclofenac on oxidative stress remains to be demonstrated.

## 4. Materials and Methods

### 4.1. Cell Culture

Huh-7, PLC-PRF-5, and HepG2 cells were cultured in Dulbecco’s Modified Eagle’s Medium-high glucose (DMEM) (Sigma-Aldrich, Buchs, Switzerland, ref. D5796) supplemented with 10% Fetal Bovine Serum (ThermoFisher Scienntific Waltham, MA, USA, ref. 10270-106) and 1% streptomycin/penicillin (BioConcept, Allschwill, Switzerland, ref. 4-01F00-H). HepG2 and PLC-PRF-5 were purchased from American Type Culture Collection (Manassas, VA USA). Huh-7 was obtained from Japanese Collection of Research Bioresources (Osaka, Japan).

### 4.2. Reagents

The following reagents were used: dimethyl sulfoxide (DMSO) (Sigma-Aldrich, St. Louis, MO, USA, ref. 41640), sorafenib (LC Laboratories, Woburn, MA, USA, ref. S-8502), diclofenac sodium salt (Sigma-Aldrich, ref. D6899), *N*-acetyl-l-cysteine (NAC) (Sigma-Aldrich, ref. A9165), *N*-acetyl-l-alanine (NAA) (Sigma-Aldrich, ref. A4625), L-ascorbic acid (AA) (Sigma-Aldrich, USA, ref. A4544), propidium iodide solution (1.0 mg/mL in water) (Sigma-Aldrich, ref. P4864), Z-VAD-FMK (Enzo Life Sciences, Lausen, Switzerland #260-020M001), necrostatin-1 (NEC-1) (Biovision, Milpitas, CA, USA, ref. 1864-5), chloroquine diphosphate salt (CQ) (Sigma-Aldrich, ref. C6628), and ethidium bromide solution (10 mg/mL in water) (Serva, Heidelberg, Germany, ref. 21251).

### 4.3. MTS Proliferation Assay

Cancer cells were plated in 96-well plates with 10,000 cells per well. After 24 h, cells were treated with the indicated concentrations of diclofenac, sorafenib, or diclofenac and sorafenib, or DMSO/water as vehicle controls, for a further 24 or 48 h. In selected experiments, indicated concentrations of *N*-acetyl-l-cysteine, *N*-acetyl-l-alanine, L-ascorbic acid, Z-VAD-FMK, necrostatin-1 or chloroquine diphosphate salt were added 1 h prior to the sorafenib and diclofenac treatment. Cellular proliferation was subsequently tested with CellTiter 96® AQueous One Solution Cell Proliferation Assay (Promega, Madison, WI, USA, ref. G3580) following the manufacturer’s instructions. Absorbance at 492 nm was measured on a microplate reader (BioRad, Cressier, Switzerland) 60 min after compound addition and expressed as a relative percentage compared to untreated control cells. Experiments were performed in triplicates and repeated 3 times.

### 4.4. Crystal Violet Staining

Cancer cells were plated in 6-well plates with 2 × 10^5^ cells per well. After 24 h, cells were treated with the indicated concentrations of diclofenac, sorafenib, or diclofenac and sorafenib, or DMSO/water as vehicle controls, for 24 h. Cells were placed on ice, washed twice with cold phosphate buffered saline (PBS), and fixed with ice cold 100% methanol for 10 min. Following treatment, cells were placed at room temperature and covered with 2 mL of Crystal Violet Solution (Sigma-Aldrich, ref. HT901-8FOZ) 0.5% solution in 25% methanol for fifteen minutes. Cells were then repeatedly washed with water and allowed to dry at room temperature for 48 h. Plates were photographed for macroscopic and microscopic visualization with a Zeiss microscope. Microscopic views were obtained with a 5× magnification lens on bright field settings and recorded with AxoVision.

### 4.5. Flow Cytometry

Cancer cells were plated in 10 cm plates with 10^6^ cells per plate. After 24 h, cells were treated with the indicated concentrations of diclofenac, sorafenib, or diclofenac and sorafenib, or DMSO/water as vehicle controls, for 24 h. For selected experiments, indicated concentrations of NAC were added 1 h prior to the sorafenib and diclofenac treatment. Floating and adherent cells were collected, washed in PBS, and fixed/permeabilized in 70% ethanol solution for 24 h at 4 °C. Cells were resuspended in PBS containing 200 µg/mL RNASe and 20 µg/mL propidium iodide and incubated at 37 °C for 30 min. DNA content was measured with a Gallios cytometer and analyzed with FlowJo V10 CL. Experiments were repeated 3 times.

### 4.6. ROS Measurement

Cancer cells were plated in 96-well plates with 1 × 10^5^ cells per well. After 24 h, cells were treated with the indicated concentrations of diclofenac, sorafenib, or diclofenac and sorafenib, or DMSO/water as vehicle controls, for 5 h. An ROS assay stain was subsequently added following the manufacturer’s instructions (Total Reactive Oxygen Species Assay Kit 520 nm (Invitrogen, Carisbad, CA, USA, ref. 88-5930-74) and incubated at 37 °C for 1 h. Total ROS levels were measured on a fluoroscopic microplate reader at 520 nm. Results are expressed as mean fluorescent intensity relative to control.

### 4.7. MitoSOX

For the creation of mitochondrial DNA-deficient (MD) cells, cancer cells were cultured for 12 weeks in the presence of 50 ng/mL ethidium bromide [18]. Both control and MD cells mediums were renewed every two days and subcultured once confluence reached 75–80%. One day prior to the experiments, cancer cells were plated in 10 cm plates at 10^6^ cells per plate. After 24 h, cells were treated with the indicated concentrations of diclofenac, sorafenib, or diclofenac and sorafenib, or DMSO/water as vehicle controls, for 12 h. Adherent cells were collected, washed once in supplemented DMEM, and 0.5 × 10^6^ cells were aliquoted in PBS. MitoSOX^™^ Red mitochondrial superoxide indicator (Invitrogen, Eugene, OR, USA, ref. M36008) at 1 µM or DMSO for control cells were subsequently added and cells were incubated in a shaking incubator at 37 °C for 30 min. Following incubation, cells were washed thrice with 0.5 mL of PBS and resuspended in a flow cytometer measurement tube in a final volume of 0.5 mL of PBS. Mean fluorescence intensity was measured with a Gallios cytometer and analyzed with FlowJo V10 CL. Experiments were performed in duplicates and repeated 3 times.

### 4.8. GSH Measurement

Cancer cells were plated in 96-well plates at 1 × 10^5^ cells per well. After 24 h, cells were treated with NAC 6 mM for 1 h and further treated with the indicated concentrations of diclofenac, sorafenib, or diclofenac and sorafenib, or DMSO/water as vehicle controls, for 5 h. Total and reduced glutathione levels were measured with the Glutathione Colorimetric Assay Kit (Biovision, Milpitas, CA, USA, ref. K261-100) following the manufacturer’s instructions. Absorbance was measured at 412 nm on a microplate reader (BioRad). Experiments were performed in duplicates and repeated 3 times. Results are expressed as nM/10^6^ cells.

### 4.9. Mouse Model

Animal experiments were performed in accordance with the Swiss federal animal regulations and approved by the local veterinary office (3051). Female, nude, eight-week old mice were purchased from Charles River. Huh-7 (1  ×  10^6^) cells, suspended in 100 µL PBS, were injected subcutaneously into the right flank. Once tumor xenografts reached ~50 mm^3^, mice were randomized into four different treatment groups (*n* = 5/group; control, diclofenac, sorafenib, sorafenib and diclofenac). Treatment consisted of diclofenac (30 µg/g mice/day/p.o), sorafenib (15 µg/g mice/day/p.o), or diclofenac in combination with sorafenib dissolved in a vehicle consisting of 75% water, 11% Ethanol 95%, 3% DMSO, and 11% Polyoxyl 35 hydrogenated castor oil (Cremophor EL, Sigma-Aldrich, ref. C5135). Treatments or control vehicles were given daily in the morning. Tumor volumes were measured each day with a caliper and estimated with the formula $V=\;\frac{\pi }{6}\times \mathrm{length}\;\times \;\mathrm{width}\;\times \mathrm{height}$. Following sacrifice of the mice, tumors were excised, measured, and weighed, and samples were processed for immunohistochemical analysis.

### 4.10. Statistics

The level of significance was determined by a two-way ANOVA with Sidak’s multicomparison test using GraphPad Prism version 8.0.1. QQ plots were used to confirm Gaussian distribution. Significance was derived from adjusted *p* values (* < 0.05, ** < 0.01, *** < 0.001, **** < 0.0001).

## 5. Conclusions

In summary, our study provides evidence of the advantages of combining diclofenac and sorafenib in HCC. More broadly, it also suggests that combining sorafenib with molecules that increase oxidative stress represents a treatment strategy that improves the efficacy of sorafenib. Nevertheless, before translating such treatment strategy into clinical trials, additional experiments are needed to fully identify the mechanisms responsible for diclofenac/sorafenib-mediated cell death. In addition, patients should be carefully selected and monitored, as concerns are raised regarding the eventual toxicity of such therapy.

## Figures and Tables

**Figure 1 cancers-11-01453-f001:**
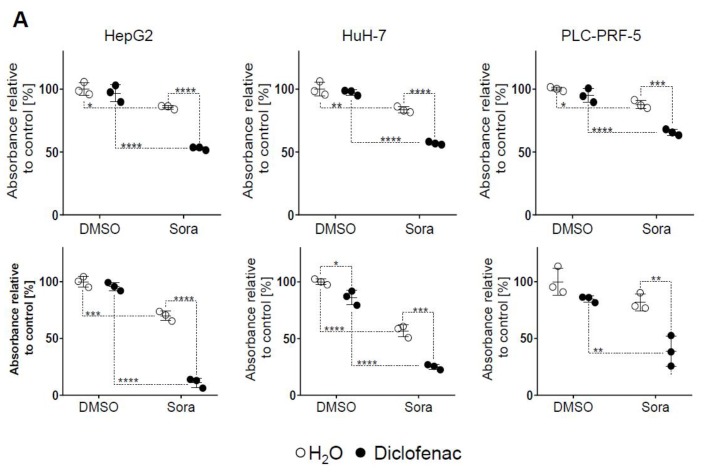
Effect of sorafenib and diclofenac on hepatocellular carcinoma (HCC) cell growth. (**A**) MTS proliferation assay of HepG2, Huh-7, and PLC-PRF-5 cell lines treated for 24 h (upper panels) or 48 h (lower panels) with sorafenib (Sora, 5 μM), diclofenac (100 μM), sorafenib and diclofenac, or dimethyl sulfoxide (DMSO) as a control for sorafenib and H_2_O for diclofenac. Each dot represents the mean of a separate experiment run in triplicates. The mean of the control group (DMSO/H_2_O) was fixed at 100%. (**B**) Photographs (5×) of HepG2, Huh-7, and PLC-PRF-5 subjected to sorafenib (Sora, 5 μM), diclofenac (100 μM), sorafenib and diclofenac, or DMSO and H_2_O treatments for 24 h. (**C**) HCC cells were treated with sorafenib (5 μM), diclofenac (Diclo, 100 μM), sorafenib and diclofenac, or DMSO or H_2_O as controls. After 24 h of treatment, cells were collected and processed for cell cycle analysis. Mean DNA content profiles of three independent experiments are shown. *p* value: * < 0.05, ** < 0.01, *** < 0.001, **** < 0.0001, two-way ANOVA with Sidak’s multiple comparisons test. For panel (**c**), *p* values were determined for the hypodiploid fractions.

**Figure 2 cancers-11-01453-f002:**
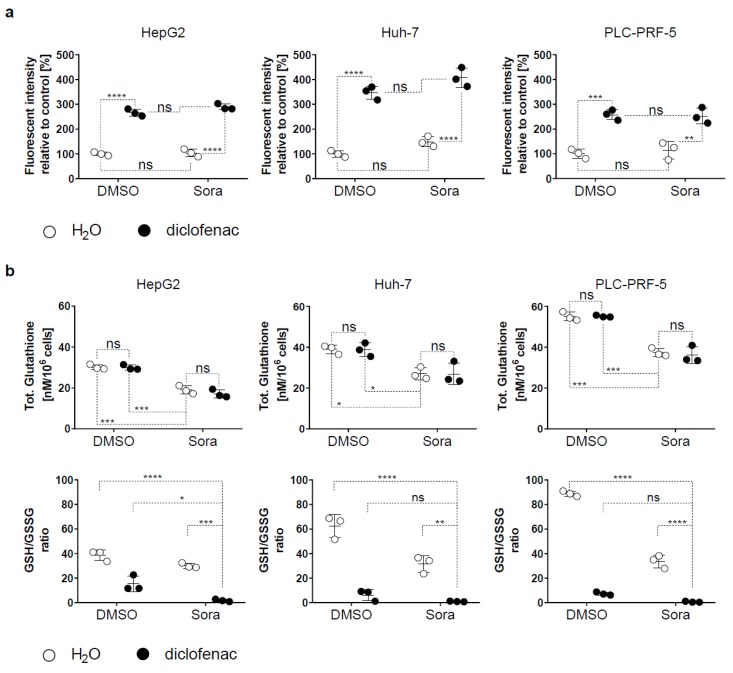
Diclofenac/sorafenib co-therapy increases oxidative stress in HCC cell lines. (**a**) HCC cells were treated with sorafenib (Sora, 5 μM) or diclofenac (100 μM), or DMSO or H_2_O as controls, for 5 h. ROS levels were determined and expressed as mean fluorescent intensity relative to control (DMSO/H_2_O treated cells). Each point represents the mean intensity of one independent experiment run in duplicates. (**b**) HCC cells were treated with sorafenib (Sora, 5 μM) or diclofenac (100 μM), or DMSO or H_2_O as controls, for 5 h. The total glutathione (upper panels) and the ratio of reduced glutathione to oxidized glutathione (GSH/GSSG ratio, lower panels) were quantified. Each dot represents the mean of an independent experiment run in duplicates. *p* value: * < 0.05, ** < 0.01, *** < 0.001, **** < 0.0001, ns: nonsignificant as indicated by a two-way ANOVA with Sidak’s multiple comparisons test.

**Figure 3 cancers-11-01453-f003:**
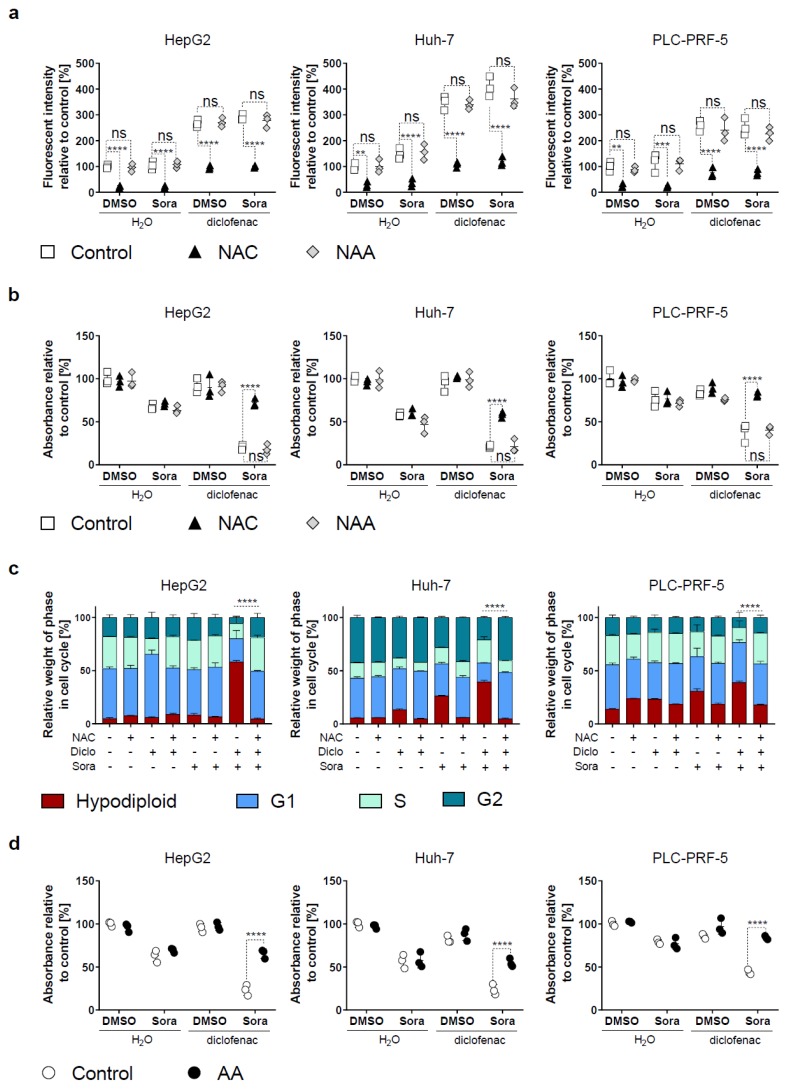
Sorafenib/diclofenac-induced HCC cell death is prevented by anti-oxidants. (**a**) HCC cells were treated with sorafenib (Sora, 5 μM) or diclofenac (100 μM), or DMSO or H_2_O as controls, for 5 h in the presence or absence of N-acetyl-cysteine (6 mM, NAC) or N-acetyl-alanine (6 mM, NAA). ROS levels were determined and expressed as mean fluorescent intensity relative to control (DMSO/H_2_O treated cells). The mean of the control condition was fixed at 100%. Each point represents the mean intensity of one independent experiment run in duplicates. (**b**) MTS proliferation assay of HepG2, Huh-7, and PLC-PRF-5 cell lines treated for 48 hours with sorafenib (Sora, 5 μM), diclofenac (100 μM), sorafenib and diclofenac, or DMSO or H_2_O as controls in the presence or absence of N-acetyl-cysteine (6 mM, NAC) or N-acetyl-alanine (6 mM, NAA). Each dot represents the mean of a separate experiment run in triplicates. The mean of the control group (DMSO) was fixed to 100%. (**c**) HCC cells were treated with sorafenib (Sora, 5 μM), diclofenac (Diclo, 100 μM), sorafenib and diclofenac, or DMSO or H_2_O as controls in presence or absence of N-acetyl-cysteine (6 mM, NAC). After 24 h of treatment, cells were collected and processed for cell cycle analysis. The mean DNA content profile of three independent experiments is shown. (**d**) MTS proliferation assay of HCC cell lines treated for 48 h with sorafenib (Sora, 5 μM), diclofenac (100 μM), sorafenib and diclofenac, or DMSO or H_2_O as controls in the presence or absence of ascorbic acid (AA, 2 mM). Each dot represents the mean of a separate experiment run in triplicates. The mean of the control group (DMSO/H_2_O) was fixed to 100%. *p* value: * < 0.05, ** < 0.01, *** < 0.001, **** < 0.0001, ns: nonsignificant as indicated, two-way ANOVA with Sidak’s multiple comparisons test**.** For panel (**c**), *p* values were determined for the hypodiploid fractions.

**Figure 4 cancers-11-01453-f004:**
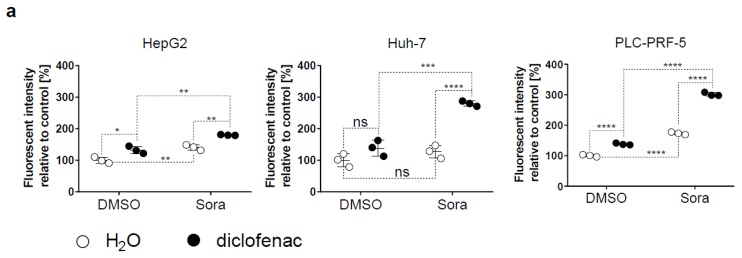
Sorafenib/diclofenac cotreatment increases mitochondrial ROS levels. (**a**) HCC cells were treated with sorafenib (Sora, 5 μM) or diclofenac (100 μM), or DMSO or H_2_O as controls, for 12 h. Mitochondrial ROS levels were quantified and expressed as mean fluorescent intensity relative to control (DMSO/H_2_O treated cells). Each point represents the mean intensity of one independent experiment run in triplicates. The mean of the control condition was set at 100%. (**b**) Huh-7 and mitochondrial DNA-deficient Huh-7 (Huh-7 MD) were treated with sorafenib (Sora, 5 μM) or diclofenac (100 μM), or DMSO or H_2_O as controls, for 12 h. Mitochondrial ROS levels were quantified. (**c**) MTS proliferation assay of Huh-7 and mitochondrial DNA-deficient Huh-7 (Huh-7 MD) treated for 48 h with sorafenib (Sora, 5 μM), diclofenac (100 μM), sorafenib and diclofenac, or DMSO or H_2_O as controls. Each dot represents the mean of a separate experiment run in triplicates. The mean of the control group (DMSO/H_2_O) was fixed at 100%. *p* value: * < 0.05, ** < 0.01, *** < 0.001, **** < 0.0001, ns: nonsignificant as indicated, two-way ANOVA with Sidak’s multiple comparisons test.

**Figure 5 cancers-11-01453-f005:**
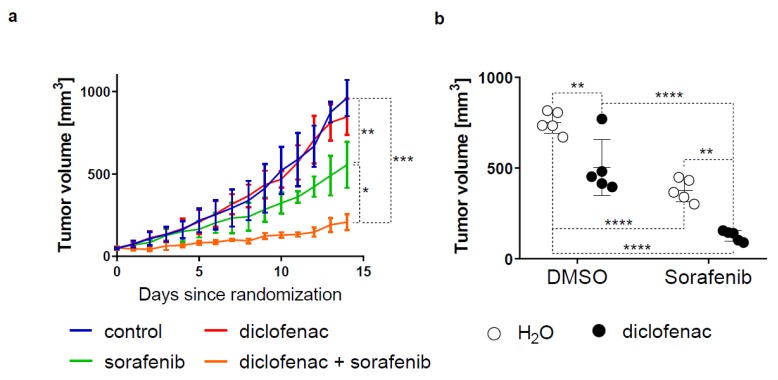
Anticancer efficacy of sorafenib and diclofenac in Huh-7 tumor xenografts. (**a**) Growth curves of Huh-7 tumor xenografts treated with the vehicle, diclofenac (30 µg/g mice/day p.o), sorafenib (15 µg/g mice/day p.o) or sorafenib and diclofenac (5 mice per group). (**b**) Tumor volumes measured ex vivo following sacrifice of the mice. (**c**) Tumor weights were determined following sacrifice of the mice. (**d**) Mice weights measured before sacrifice. *p* value: * < 0.05, ** < 0.01, *** < 0.001, **** < 0.0001, ns: nonsignificant, two-way ANOVA with Sidak’s multiple comparisons test.

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
