# Peer review of "Diclofenac Potentiates Sorafenib-Based Treatments of Hepatocellular Carcinoma by Enhancing Oxidative Stress"

_cancers, 2019, doi:10.3390/cancers11101453_

Round 1
Reviewer 1 Report
I read with pleasure this paper about the possible synergistic effects of sorafenib and diclofenac in the treatment of HCC. The methods are elaborate and the results clearly presented. As such, I have no specific comments about the EFFICACY of this combination which the Authors clearly demonstrated both in vitro and in a murine model.
I have, however, a relevant observation about the SAFETY of such a strategy. First, sorafenib is ussulally prescribed until disease progression or intolerance: ideally, it is a long-term medication since its discontinuation its an harbinger of a short survival. In this setting, diclofenac should be used as a long-term medication as well. The gastrointestinal adverse events, however, limit this possibility. Moreover, most HCC patients also suffer from liver cirrhosis (80-90%). In this setting, NSAIDs are particularly hazardous as they may cause hepatorenal syndrome, ascitic decompensation, or even a direct liver injury.
The safety aspect, however, has been completely neglected in the manuscript. instead, the Authors should consider this problem that may severely affect the clinical relevance of their findings.
Author Response
We thank reviewer 1 for his/her positive comments and suggestions. We fully agree that the toxicity of a diclofenac/sorafenib regimen was neglected. We therefore have added the following paragraph in the discussion section:
Translating our treatment regimen to patients will require careful assessments of drug toxicities. Indeed hepatotoxicity due to NSAID-induced hepatocellular injury has been well documented {Bessone, 2010 #39}. Although the incidence of liver injury induced by NSAID remains low, NSAID could be particularly detrimental in patients with HCC as they frequently present liver cirrhosis. In addition, NSAID could precipitate renal failure and increase risk of bleeding in these patients. Besides its own liver toxicity, diclofenac interaction with sorafenib might further induce side effects. In fact, metabolism of diclofenac might be slowed down via sorafenib-mediated inhibition of cytochrome P450 2C9 metabolic pathway of diclofenac {Van Booven, 2010 #44}{Pajares, 2012 #42}. Consistent with this, acute liver failure was reported in a patient suffering from kidney cancer and treated simultaneously with sorafenib and diclofenac {Yin, 2019 #46}. Hence, caution has to be taken before translating such treatment strategy in the clinic and should be initially tested in HCC patients without cirrhosis to minimize the risk to develop toxic side effects.
Reviewer 2 Report
There are several reports that phosphorylated-c-Jun contributes to sorafenib-resistance.
ex) Haga Y, et al. Overexpression of c-Jun contributes to sorafenib resistance in human hepatoma cell lines. PLoS One. 2017 Mar 21;12(3):e0174153. doi: 10.1371/journal.pone.0174153.
Authors should examine them and discuss more.
Author Response
We thank reviewer 2 for his/her comments. As suggested we added this pragraph in the discussion section regarding c-jun
Like for other targeted therapies, development of resistance is a major obstacle to the anti-cancer benefit of sorafenib {Zhu, 2017 #6}. In this context, overexpression and activation of the transcription factor c-jun plays an important role in contributing to sorafenib resistance {Haga, 2017 #41}{Chen, 2016 #40}{Xiang, 2019 #45}. Indeed, activation of c-jun is enhanced by sorafenib in HCC cell lines and inhibition of c-jun increases sorafenib-mediated apoptosis {Haga, 2017 #41}. Importantly, in HCC patients, tumors with high levels of c-jun phosphorylation are less sensitive to sorafenib compared to tumors with low levels {Chen, 2016 #40}. Therefore, it will be important to test in future experiments our treatment strategy in context of sorafenib resistance and particularly address the effects of sorafenib/diclofenac on c-jun activity. Of note, it was reported that diclofenac increases c-jun expression in acute myeloid leukemia cell lines {Singh, 2011 #43}.
Reviewer 3 Report
Authors of the manuscript entitled „Diclofenac potentiates sorafenib-based treatments of hepatocellular carcinoma by enhancing oxidative stress” present an interesting topic in the field of HCC carcinoma therapy, but manuscript should be corrected before being accepted for publication.
General
Authors should correct the order of the manuscript sections (Research manuscript sections: Introduction, Materials and methods, Results, Discussion).
In accordance with the requirements: “Abbreviations should be defined in parentheses the first time they appear in the abstract, main text, and in figure or table captions and used consistently thereafter.” The abbreviations, for example: RET, NAC, DMSO, DMEM, has not been clarified.
Abstract
Abstract should be improved. Abstract should be a total of about 200 words maximum. Currently abstract is too long with excessive wording (256 words). Authors wrote that: “Anti-oxidant compounds including N-acetyl-cysteine or vitamin C reversed the deleterious effects of diclofenac/sorafenib co-therapy suggesting that generation of toxic levels of oxidative stress was responsible for cell death.” Does it concern own research? There are no information about N-acetyl-cysteine and vitamin C in the manuscript, so I suggest either to expand this part of text in another section of the manuscript, or to remove this information from the Abstract. In my opinion, Authors should present in the abstract information about the dose and duration of diclofenac/sorafenib use.Keywords
Authors proposed only 4 keywords. Authors should add new keywords specific for this article.
Introduction
Authors should provide more information on the current state of knowledge on the use of diclofenac and sorafenib/diclofenac co-therapy in hepatolcellular carcinoma. Are there any studies involving patients with HCC? Better justification is needed for the study.
Materials and methods
For the ethical issues Authors should precisely indicate number of ethical commission agreement. Authors should complete the information about the statistical tests that have been applied. How did Authors verify the normality of distribution of data? What tests did they use? In accordance with the requirements: “Methods sections for submissions reporting on research with cell lines should state the origin of any cell lines. For established cell lines the provenance should be stated and references must also be given to either a published paper or to a commercial source”. Authors should provide adequate information about the applied cell lines. What are the units of the following parameters: ROS level, total and reduced glutathione levels? It should be completed in this section. Authors should also add more information about the experiment on mice and precisely specify the time of administration sorafenib/diclofenac.Results
The figure 1b is hardly visible. Authors, should correct the quality of the photography Discussion
I suggest to use additional literature on HCC therapy: Yagi et al. Biol Pharm Bull. 2014;37(7):1234-40. Di Giacomo et al. Arch Toxicol. 2019 Mar;93(3):623-634. doi: 10.1007/s00204-019-02395-9. The discussion should be improved. First, Authors should discuss own results in relation to the work of other authors, and then provide conclusions from own research. There is no information about doses of sorafenib used in other studies. Authors should precisely describe the results from other experiments. Authors should also present the limitations of their study. Moreover, Authors may provide also information about the possible future research directions.Conclusions
The conclusions are insufficient. Is it safe to use diclofenac/sorafenib co-therapy in patients with HCC? What are the possible side effects?
Authors of the manuscript entitled „Diclofenac potentiates sorafenib-based treatments of hepatocellular carcinoma by enhancing oxidative stress” present an interesting topic in the field of HCC carcinoma therapy, but manuscript should be corrected before being accepted for publication.
Author Response
We thank reviewer 3 for his/her comments. We have followed his/her recommandations to imrpove the quality of the manuscript as follows:
General:
We have kept the original structure of our manuscript as according to instructions for authors of cancers journal (https://www.mdpi.com/journal/cancers/instructions) the order of manuscript sections should be as follow: introduction, Results, discussion, materials and methods, conclusions.
We added the full names that were missing to the following abbreviations: fms-like tyrosine kinase 3 (flt-3); rearranged during transfection kinase (RET); N-acetyl-cysteine (NAC); N-acetyl-alanine (NAA); Dulbecco's Modified Eagle's Medium (DMEM), dimethyl sulfoxide (DMSO)
Abstract:
We have shortened the abstract that contains now 202 words. N-acetyl-cysteine (NAC) and vitamin C refer to figure 3. However, confusingly in figure 3, the chemical name of vitamin C, ascorbic acid (AA) was used throughout the text. We therefore replaced vitamin C by ascorbic acid in the abstract. We have removed some methodological aspects of the abstract that we thought less relevant than the rest. As suggested we also added the concentrations and durations of treatments
Keywords:
Cell death and co-therapy were added as keywords
Introduction:
To date diclofenac was tested only in HepG2 cells in vitro and showed like other NSAID anti-proliferative effects. Currently no study is being conducted in patients. The anti-cancer potential of sorafenib/diclofenac was also identified in melanoma cells hence justifying its investigation in HCC. We added accordingly one sentence in the introduction
Materials and methods:
As asked the following details were added in the materials and methods section:
The number of the ethical authorization was added. We used two-way ANOVA with Sidak’s multiple comparisons test. Regarding normality of data we used QQ plots. The origin of every cell line is now stated in the text. ROS levels are expressed as mean fluorescent intensity relative to control condition. Total and reduced glutathione levels are expressed as nM/106 cells. Regarding mice experiments treatments were given daily in the morning.
Results:
We added contrast to the image which should show damages done on cell monolayer by sorafenib/diclofenac rather than cell details.
Discussion and conclusions:
We have followed the recommandations and added different chapter in the discussion that appear in red in the revised manuscript
Round 2
Reviewer 1 Report
I thank the Authors for considering my point. Their comments about safety are timely and appropriate. I have no further comments and I recommend this manuscript for publication.
Author Response
Thanks a lot for your review and for your nice comments
Reviewer 2 Report
Duval et al. studied well, however, authors should improve minor points below.
in Figures and their legends. In Figure 1C, what is diclo? Authors should write that in legends. Over all, it should be better to use "same word" in text, figures and figure legends.Author Response
Thank you for your comment
We agree with you that we should have used the same wording for diclofenac throughout the text. We have added the meaning of diclo in the legend of figure 1c
Reviewer 3 Report
Authors corrected the manuscript according to my suggestions, so I accept the manuscript in present form.
Author Response
Thank you for your review and your comments